# Four-Chlorophenoxyacetic Acid Treatment Induces the Defense Resistance of Rice to White-Backed Planthopper *Sogatella furcifera*

**DOI:** 10.3390/ijms242115722

**Published:** 2023-10-29

**Authors:** Wanwan Wang, Haiyun Rui, Lei Yu, Nuo Jin, Wan Liu, Chen Guo, Yumeng Cheng, Yonggen Lou

**Affiliations:** 1Jiangsu Key Laboratory of Chiral Pharmaceuticals Biosynthesis, Taizhou University, Taizhou 225300, China; ruihaiyun@tzu.edu.cn (H.R.); yulei2637@163.com (L.Y.); liuwan_0802@163.com (W.L.); guochen427724@163.com (C.G.); cym5201@yeah.net (Y.C.); 2State Key Laboratory of Rice Biology & Ministry of Agriculture Key Lab of Molecular Biology of Crop Pathogens and Insects, Institute of Insect Sciences, Zhejiang University, Hangzhou 310058, China; njin@zju.edu.cn

**Keywords:** chemical elicitor, 4-chlorophenoxyacetic acid, lignin-like polymers, white-backed planthoppers, rice, chemical defense

## Abstract

Chemical elicitors can increase plant defense against herbivorous insects and pathogens. The use of synthetic chemical elicitors is likely to be an alternative to traditional pesticides for crop pest control. However, only a few synthetic chemicals are reported to protect plants by regulating signaling pathways, increasing the levels of defense metabolites and interfering with insect feeding. Here, we found that the exogenous application of a phenoxycarboxylic compound, 4-chlorophenoxyacetic acid (4-CPA), can induce chemical defenses to protect rice plants from white-backed planthoppers (WBPH, *Sogatella furcifera*). Four-CPA was rapidly taken up by plant roots and degraded to 4-chlorophenol (4-CP). Four-CPA treatment modulated the activity of peroxidase (POD) and directly induced the deposition of lignin-like polymers using hydrogen peroxide (H_2_O_2_) as the electron acceptor. The polymers, which are thought to prevent the planthopper’s stylet from reaching the phloem, were broken down by WBPH nymphs. Meanwhile, 4-CPA increased the levels of flavonoids and phenolamines (PAs). The increased flavonoids and PAs, together with the degradation product of the polymers, avoided nymphal feeding and prolonged the nymphal period for 1 day. These results indicate that 4-CPA has the potential to be used as a chemical elicitor to protect rice from planthoppers. Moreover, these findings also open a pathway for molecule structure design of phenoxycarboxylic compounds as chemical elicitors.

## 1. Introduction

Plants, when damaged by herbivorous insects, will recognize herbivore-associated signals and then produce specific defense responses [1,2,3]. Thus far, many herbivore-associated signals, such as volicitin, inceptin, caeliferins, bruchins, and β-glucosidase have been identified [4,5,6,7,8]. Furthermore, studies on the interaction mechanisms underlying how herbivores induce plant defense responses have revealed that multiple signaling pathways are involved in these defense responses in plants. These pathways mainly include defense-related pathways mediated by jasmonic acid (JA), salicylic acid (SA), and ethylene. The modulation of these defenses also involves pathways mediated by gibberellins, brassinosteroids, auxins, and cytokinins [9,10,11,12]. Crosstalk among these signaling pathways causes large changes in transcriptomes, proteomes, and metabolomes in plants, and these changes may reduce the palatability of plants to herbivores and increase the attractiveness of plants to natural enemies of herbivores, thereby enhancing the direct and/or indirect resistance of plants to herbivores [13].

To control these insect pests, farmers often extensively utilize chemical insecticides. However, chemical pesticides are harmful to crops, the environment, and human health. Therefore, alternative insect pest management is required. Exploitation of the induced resistance stated as above, elicited by chemical elicitors, is one attractive alternative [14,15]. It has been reported that the use of some hormones and analogues of plant origins, oral secretory proteins of animal origins, and inorganic ions can help plants to defend themselves against herbivorous insects [14,16,17]. JA and its analogues, such as methyl jasmonate (MeJA), coronalon, jasmonoyl-isoleucine (JA-Ile), and other JA–amino acid conjugates have been the most studied in this regard. These chemicals can induce the biosynthesis of various direct and indirect defensive compounds in plants, thereby helping protect plants from herbivorous insects [14,18,19,20]. In addition, many synthetic chemicals have been found to have similar biological activities [15,19,21,22]. Most of the resistance mechanisms are based on plant chemical defenses, such as increased levels of volatiles, phenolic polymers, and protease inhibitors [14,15,22,23]. Compared to plant-derived elicitors, synthetic chemicals have greater potential for use in plant insect resistance due to their ease of access and affordability.

Rice, one of the world’s most important food crops, suffers from herbivores from the seedling to the mature stage. Planthoppers, including brown planthopper (BPH) *Nilaparvata lugens* (Stål), white-backed planthopper (WBPH) *Sogatella furcifera* (Horvath), and small brown planthopper (SBPH) *Laodelphax striatellus* (Fallén), are small sap-sucking insects that live on the phloem sap of rice plants. They can also damage plants by spreading virus disease. If nothing is done, the annual loss in rice yield caused by these planthoppers can reach 40% [24]. Synthetic chemical elicitors have been considered as an alternative to traditional pesticides for planthopper control. Among these synthetic chemicals, phenoxycarboxylic compounds are attracting the attention of scientists due to their low cost of synthesis and relative ease of access [15,22]. A well-known phenoxyacetic acid herbicide, 2,4-Dichlorophenoxyacetic acid (2,4-D) was reported to act as potent inducer of rice defenses by directly reducing herbivore performance or indirectly attracting the natural enemies of herbivores [22]. Four-fluorophenoxyacetic acid (4-FPA) induces the formation of phenolic polymers that inhibit the WBPH stylets from reaching the phloem sap [15]. Based on these results, we aim to determine if there are any other phenoxycarboxylic compounds that can be used as chemical elicitors. We chose four-chlorophenoxyacetic acid (4-CPA) as a research subject because it (1) has a similar structure to 4-FPA, and (2) it can be rapidly taken up by plant roots. Will 4-CPA be metabolized after uptake by the plant, and will it affect the plant’s resistance to herbivorous insects? We hypothesized that 4-CPA, like 4-FPA, would induce the deposition of phenolic polymers. Histochemical staining was used to determine whether phenolic polymers were produced and whether they were degraded by certain factors. In addition to this, we also detected changes in many secondary metabolites associated with insect resistance in plants. Our aim was to determine whether 4-CPA can act as chemical elicitor and, if so, to explore its specific mechanisms of action.

## 2. Results

### 2.1. Four-CPA Treatment Affects Plant Growth

Exogenous application of 4-CPA inhibited plant root growth (Figure 1A,B). After treatment with 4-CPA (0, 2, 5, 10 mg L^−1^) for 10 days, the plants showed significantly lower root masses compared to the control plants, while there was no obvious difference among the different concentration of 4-CPA (Figure 1B). However, the 4-CPA treatments had no effect on the length of the roots, even at a concentration of 10 mg L^−1^ (Figure 1C). We also tested the stem strength of the plants and found that the stem strength could be significantly increased through 4-CPA treatment, reaching 11.4 N for 5 mg L^−1^ and 13.6 N for 10 mg L^−1^ (Figure 1D). But, there was no difference in stem strength between the control and 2 mg L^−1^ 4-CPA plants. Based on these results, we chose 5 mg L^−1^ as the working concentration.

### 2.2. Four-CPA Is Converted into 4-Chlorophenol in Plants

In order to understand the metabolic pathway of 4-CPA in plants, we studied the changes in the levels of 4-CPA and its possible metabolite 4-chlorophenol (4-CP) in plants after treatment with 4-CPA for 1 d. Liquid chromatography–mass spectrometry/mass spectrometry (HPLC-MS/MS, Agilent 1260/6460) was used to determine the levels of 4-CPA and 4-CP (Figure 2C,D); see Appendix A for amplified mass spectra of 4-CPA (Appendix A) and 4-CP (Appendix A). Four-CPA could be quickly absorbed by the plant roots, and its content reached 3.78 µg g^−1^ on the second day after treatment (Figure 2A). The 4-CPA was then immediately degraded to 4-CP, and its content reduced to the lower limit of detection (Figure 2A). On day 4 of the 4-CPA treatment, 4-CP was found to peak at 6.13 µg g^−1^ and then declined rapidly, reaching only 14.83% of its peak on day 7 (Figure 2B). In order to test whether 4-CPA and 4-CP are toxic to WBPH, survival experiments were carried out. As shown in Appendix A, the addition of 4-CPA and 4-CP to the artificial diet had no effect on WBPH survival.

### 2.3. Four-CPA Treatment Triggers Transcriptional Changes in Plants

In order to understand the response mechanism of 4-CPA in rice, plants were treated with 0 (control; Con) or 5 mg L^−1^ of 4-CPA (Figure 3A) to analyze differentially expressed genes (DEGs) at the level of the transcriptome using RNA-sequencing (RNA-Seq). DEGs were identified based on a *p* value ≤ 0.05. Transcriptome analysis demonstrated that 2841 of 5736 DEGs were upregulated and 2895 were downregulated in response to the 4-CPA treatment (Figure 3B). We validated these DEGsusing qRT-PCR and found that all of the 15 selected genes were in agreement with the transcriptome data (Appendix A). The heatmap confirmed the difference in expression patterns between the control and 4-CPA treatments (Figure 3C). KEGG enrichment analysis was performed to further investigate the biological functions of the DEGs identified under the 4-CPA treatment. These DEGs were enriched in 107 KEGG pathways. Figure 3D shows the top 20 KEGG enrichment pathways of the DEGs upregulated by 4-CPA. Among these pathways, 18 of the 20 pathways were related to metabolism. The phenylpropanoid biosynthesis pathway has 41 upregulated DEGs, which is one of the metabolic pathways with the highest number of upregulated DEGs (Figure 3D).

### 2.4. Four-CPA Alters Levels of H_2_O_2_ and Activity of POD

The transcriptomic data showed that 4-CPA altered the expression levels of genes in the hydrogen peroxide (H_2_O_2_)-generating and scavenging system. A total of 20 out of the 33 DEGs that generate H_2_O_2_ were upregulated (Figure 4D), and 54 out of the 77 DEGs that scavenge H_2_O_2_ were upregulated (Figure 4A,E). Most of the H_2_O_2_ scavenging DEGs were peroxidases (PODs). Only 13 of the 56 POD DEGs were downregulated, and the rest were all upregulated (Figure 4A). POD can utilize phenolic compounds as electron donors and H_2_O_2_ as an electron acceptor to produce phenolic polymers (Figure 4B). Therefore, the activity of POD and the levels of H_2_O_2_ in the plants treated with 4-CPA were investigated. Consistent with the transcriptomic data, the activity of POD increased in 4-CPA-treated plants (Figure 4C), and the increased POD activity was not affected by WBPH nymph infestation (Figure 4C). H_2_O_2_ levels were found to be reduced in both 4-CPA-treated and 4-CPA + WBPH-infested plants (Figure 4F).

### 2.5. WBPH Inhibits 4-CPA-Induced Deposition of Phenolic Polymers

Phenolics, as substrates for POD, can polymerize to form phenolic polymers using H_2_O_2_ as the electron acceptor. 4-CP, the degradation product of 4-CPA, is a typical phenolic compound. As we observed a decrease in 4-CP and H_2_O_2_ levels together with an increase in POD activity in the 4-CPA-treated plants, we hypothesized that there would be phenolic polymers deposited in the rice. Laser confocal images of the rice leaf sheaths confirmed our hypothesis. It was found that there were many polymer particles deposited in the parenchyma cells of the leaf sheaths 4 d post-4-CPA application (Figure 5A, second panel). The polymers were not observed in the parenchyma cells of the leaf sheath treated with 4-CPA and simultaneously infested with WBPH nymphs (Figure 5A, third panel). To find out if these formed polymers were degraded by WBPH nymphs, the plants were treated with 4-CPA for 4 days, when polymers had formed, before infestation with WBPH nymphs. The result showed that the 4-CPA-induced polymers disappeared at the site where the WBPH nymphs fed on the fourth day of feeding (Figure 5A, fourth panel).

The deposition of phenolic polymers was considered to be associated with the penetration of the WBPH stylet into the phloem cells [15]. With the disappearance of polymers, WBPH nymphs were hypothesized to survive normally. The biochemical experiment on the survival rate of planthoppers proves our hypothesis. The results showed that 4-CPA did not affect the survival of WBPH nymphs, whether the nymphs were placed on the plants for infestation before (Figure 5B) or after (Figure 5C) the phenolic polymers formed.

### 2.6. Four-CPA Increases the Expression Levels of DEGs Involved in the Phenylpropanoid Biosynthesis Pathway

The results of the KEGG enrichment analysis showed that the phenylpropanoid biosynthesis pathway was the metabolic pathway with the highest number of upregulated genes (Figure 3D). Therefore, the DEGs of the phenylpropanoid biosynthesis pathway were analyzed comprehensively. Figure 6 presents a chart of the phenylpropanoid biosynthesis pathway in plants, showing that genes encoding phenylalanine ammonia-lyase (PAL), trans-cinnamate 4-monooxygenase (C4H), shikimate O-hydroxycinnamoyltransferase (HCT), caffeoyl-CoA O-methyltransferas (CCOAOMT), coniferyl-aldehyde dehydrogenase (HCALDH), and ferulate-5-hydroxylase (F5H) were all significantly upregulated by 4-CPA. Only one of the five genes encoding cinnamoyl-CoA reductase (CCR), one of the five genes encoding 4-coumarate-CoA ligase (4CL), one of the four genes encoding cinnamyl-alcohol dehydrogenase (CAD), and three of the twenty-one genes encoding peroxidase (POD) were downregulated, while the rest were upregulated (Figure 6).

### 2.7. Four-CPA Increases Levels of Flavonoids and Phenolamines

Many secondary metabolites from the phenylpropanoid pathway, such as flavonoids and phenolamines (PAs), play a crucial role in plant resistance to insects [25,26,27,28]. Since most of the genes involved in the phenylpropanoid biosynthesis pathway were upregulated by the 4-CPA treatment, we investigated the levels of flavonoids and PAs. We found that five of the fifteen flavonoids were increased following the 4-CPA treatment, and ten were increased following the 4-CPA+ WBPH treatment (Figure 7A). In terms of PAs, 10 out of the 13 PAs had significantly higher levels in the 4-CPA-treated plants than in the control plants (Figure 7B). As lignin is the main product of the phenylpropanoid biosynthesis pathway, we also measured the lignin contents. The result showed that the 4-FPA treatment had no effect on lignin levels (Appendix A).

### 2.8. Four-CPA Treatment Influences the Performance of WBPH Nymphs

Although the 4-CPA treatment did not reduce the survival rate of WBPH nymphs, it is not known whether changes in flavonoids and phenolamines (PAs) have an effect on other performances of WBPH. Here, we investigated the feeding preference and development time of nymphs. Our study showed that the WBPH nymphs preferred to feed on the control plants compared to the plants that had been grown in a 4-CPA-treated nutrient solution for 4 d (Figure 8A). And the development time of the WBPH nymphs that fed on 4-CPA-treated plants was longer than that of the nymphs that fed on the control plants (Figure 8B).

## 3. Discussion

The use of chemical elicitors is likely to be an alternative to traditional pesticides for crop pest control. Unlike conventional insecticides, which act directly on insects, chemical elicitors are used to reduce plant damage from herbivorous insects by inducing morphological changes, activating resistance-related signaling pathways, and altering levels of relevant defense compounds in plants [16,29,30,31]. MeJA, *cis*-jasmonone, benzo-(1,2,3)-thiadiazole-7-carbothioic acid S-methyl ester (BTH), 2,4-D, and 4-FPA have been reported to increase plant resistance to herbivorous insects in the field [14,15,22,32]. With limited material and energy, induced defense in plants often brings about growth inhibition. As a plant growth regulator, 4-CPA significantly inhibited the growth of rice roots at a concentration ranging from 2 to 10 mg L^−1^ when applied to the roots of rice seedlings grown for 30 days in a greenhouse. This is consistent with the effects of two other phenoxycarboxylic compounds, 4-FPA and 2,4-D, on the roots of rice grown in a greenhouse [15,33]. This significant inhibitory effect on rice roots could affect the practical application of 4-CPA as a chemical elicitor. However, the age of plants, growing conditions, and application dosage and method of chemicals can all influence chemical uptake and translocation. The field environment is more complex, and the inhibitory effects of chemicals on plant growth can be attenuated under suitable growing conditions. We found no difference in the root growth of 4-FPA-treated and control rice in the field, with a control effect of 72% against WBPH nymphs [15]. As a phenoxycarboxylic compound similar to 4-FPA, 4-CPA will need to be tested in further field trials to see if it gives the same results. In addition to changes in the growth phenotypes of roots, the stem strength of 4-CPA-treated plants increased. This was due to the accumulation of granular polymers in the parenchyma cells of the leaf sheath. These changes suggest that 4-CPA could be taken up and translocated by rice plants. In fact, we did find 4-CPA in the rice stems.

Chemical elicitors can protect plants from pests and pathogens. Most elicitors exert their induced resistance effects on insects by modulating stress hormone signaling pathways, especially the JA pathway [16,29,34]. Here, we report on a chemical elicitor, 4-CPA, which enhances plant insect resistance. In this study, we found that the expression levels of most of PODs and H_2_O_2_-generating genes were increased. POD activity increased significantly after 4-CPA treatment, while the H_2_O_2_ content in the plants decreased. Furthermore, a large number of 4-CPA-induced phenolic polymers were deposited in the parenchyma cells of the leaf sheath. A substantial number of published studies demonstrate that the polymerization of phenolics is catalyzed by POD using H_2_O_2_ as the electron acceptor [35,36,37,38,39]. The presence of similar substances was reported by Graham et al. in 1991 [40]. They found that POD activity was significantly increased and phenolic polymers were rapidly accumulated in soybean cotyledon cells after treatment with a cell wall glucan elicitor from *Phytophthora megasperma*. Based on the above research, we speculate that POD catalyzed the polymerization of phenolics using H_2_O_2_ as the electron acceptor in the plant after 4-CPA treatment. Our findings were consistent with a previous study from our research group. Using H_2_O_2_ as the electron acceptor, 4-FPA, could induce the deposition of phenolic polymers formed by POD catalyzing 4-fluorophenol (the degradation products of 4-FPA) and flavonoids [15]. The phenolic polymers were not degraded by WBPH nymph feeding and were thought to prevent the stylet of planthoppers from reaching the phloem, thereby impairing feed intake and significantly reducing the survival of WBPH nymphs [15]. Interestingly, WBPH nymphs could break down the 4-CPA-induced polymers, and their survival was no longer affected. We speculate that the exact chemical structures of 4-CPA- or 4-FPA-induced phenolic polymer are the most likely reason for the discrepancies in the results between the two studies. The reasons are as follows: (i) fluorine is more electronegative than chlorine, and the C-F bond has a higher bond energy than the C-Cl bond, making the overall polymer structure more stable [41]; (ii) flavonoids were involved in the 4-FPA-induced polymer formation, but no depletion of flavonoids was detected in the 4-CPA-treated plants. Additionally, what are the degradation products of 4-CPA-induced polymers after WBPH infestation? Given the complexity of metabolites in plants, it is difficult to identify the chlorine-containing products of polymer degradation without the use of isotopic tracer techniques [42,43]. Therefore, further research and analysis is needed on this topic.

Plants depend on chemical defenses as one of the most important strategies to protect themselves from herbivore attacks. As one of the major secondary metabolic pathways in plants, phenylpropanoid biosynthesis, such as flavonoids, PAs, anthocyanins, and organic acids, can function as feeding deterrents, digestibility reducers, and toxins in regulating the interactions between plants and herbivorous insects [44,45,46,47]. For example, flavonoids accumulate rapidly in rice after infestation by planthoppers [25]. Alamgir et al. found that N-feruloylputrescine and N-p-coumaroylputrescine were strongly accumulated in BPH-attacked rice leaves and had direct lethal toxicity to BPH when added to the feeding solution [27]. A caffeoylputrescine-green leaf volatile (GLV) compound that conferred nonhost resistance to *Empoasca* leafhoppers has been identified in wild tobacco plants [48]. In this study, transcriptome data showed that 4-CPA treatment significantly induced phenylpropanoid biosynthesis pathway genes. Five and ten flavonoids were found to be increased by 4-CPA treatment and 4-CPA + WBPH treatment, respectively. In addition, 4-CPA treatment significantly increased the level of 12 PAs. Among these PAs, feruloylagmatine (Fer-Agm); N-feruloylputrescine (Fer-Put); and N1, N10-diferuloylspermidine (Difer-Spd) have been shown to be directly toxic to WBPH at a concentration of 100 µg/mL, but not at 5 µg/mL [49]. Our quantitative analysis indicated that 4-CPA-treatment resulted in the induction and accumulation of flavonoids and PA defensive compounds. Although the mortality of WBPH nymphs feeding on the plant treated with 4-CPA did not change significantly, an obvious escape behavior response of WBPH nymphs to 4-CPA treatment was found, and the developmental time of nymphs that feed on 4-CPA-treated plants was significantly longer than those fed the control plants. The above analyses show that the performance of the WBPH nymphs was still affected by the 4-CPA treatment. There are two possible reasons for this: (i) most PAs in the plants were found at less than 1 μg/g, a concentration that was insufficient to cause direct mortality of WBPH. However, the increased defense metabolites enhanced plant resistance and affected the growth and development of WBPH nymphs. (ii) The degradation products of the 4-CPA-induced polymers after WBPH infestation may also have an effect on the performance of WBPH nymphs.

In summary, we propose that 4-CPA works as follows: 4-CPA undergoes rapid side-chain degradation to yield 4-CP after being absorbed by plant roots. Four-CPA treatment induces the activity of POD. POD then catalyzes 4-CP and possibly other phenolics in plants to form polymers using H_2_O_2_ as the electron acceptor. The polymers, which are thought to prevent the planthopper’s stylet from reaching the phloem, are degraded by WBPH nymphs. Meanwhile, 4-CPA increases the levels of flavonoids and PAs. The increased flavonoids and PAs, together with the degradation products of the polymers, avoid nymphal feeding and prolong the nymphal period (Figure 9). These results suggest that 4-CPA may be used as a chemical elicitor to protect rice from planthoppers. Moreover, given that the polymers induced by 4-CPA can be degraded by WBPH nymphs, while the polymers induced by 4-FPA are not degraded, these findings also open a pathway for molecule structure design of phenoxycarboxylic compounds as chemical elicitors.

## 4. Materials and Methods

### 4.1. Plant Growth and Insect Rearing

The rice genotype used in this study was Xiushui 110, which is susceptible to WBPH. The seeds were germinated and cultured in petri dishes (diameter 8 cm, height 2 cm) in a humid incubator (27 ± 2 °C, 12 h light phase, 70% relative humidity, 15,000 Lux). Seven-day-old seedlings with a uniform size were transferred into 24-L hydroponic boxes containing a rice nutrient solution. The boxes, each with 72 plants, were put in a glasshouse at 27 ± 2 °C under a 14 h light/10 h dark photoperiod. The greenhouse had a light intensity of 30,000 lux. The 30-day-old seedlings were used for the experiments. Original WBPH colonies were collected from rice fields in Hangzhou, China, and grown on TN1 rice seedlings; a WBPH susceptible variety.

### 4.2. Compounds

Four-CPA (purity, 99.5%) and 4-CP (purity, 99.5%) were purchased from Aladdin (Shanghai, China). All standard flavonoids were chromatographic grade and were purchased from BioBioPha (Kunming, China). All standard PAs were chromatographic grade and were purchased from Chemipanda (Hangzhou, China).

### 4.3. Plant Treatments

For the 4-CPA treatment, 4-CPA (previously dissolved in acetone at a concentration of 200 mg L^−1^) was appended into the nutrient solution to obtain concentrations ranging from 2 to 10 mg L^−1^. The control (Con) plants received the same amount of acetone as the treated plants. For the 4-CPA + WBPH treatment, plants were treated with 5 mg L^−1^ of 4-CPA, and each plant was simultaneously infested with 15 third-instar nymphs of WBPH. Empty cages were attached to the plants as controls. The cages were made of glass, and cage each had a height of 8 cm and a diameter of 5 cm, with 24 small holes of 0.8 mm in diameter.

### 4.4. Measurement of Plant Growth Parameters

Plants treated with 0 to 10 mg L^−1^ 4-CPA for 10 d were harvested for measurement of plant growth parameters, including root mass, root length, and stem strength. The root length was measured from the base of the stem to the longest root tip. A root cut from the base of the stem was weighed using an analytical balance and recorded as the root mass. The stem strength was measured using a plant stem strength tester (Hangzhou Lvbo, YYD-1), which could record the momentary force generated by needling. Each of these experiments was repeated 6 times.

### 4.5. Quantification of 4-CPA and 4-CP in Plants

Changes in the levels of 4-CPA and the possible metabolites were investigated to figure out the metabolic pathway of 4-CPA in the plants. The plants were treated in two ways: (1) the plants were treated with 5 mg L^−1^ 4-CPA for 1 d and then transferred to a nutrient solution without 4-CPA; and (2) the plants were treated with 5 mg L^−1^ 4-CPA, and each plant was simultaneously infested with 15 third-instar nymphs of WBPH. After 1 d, these plants were transferred to a nutrient solution without 4-CPA. The stems were harvested for quantitative analysis of 4-CPA and its possible metabolites at 0, 2, 4, and 7 d after starting the 4-CPA treatment. Each sample was ground in liquid nitrogen and weighed to the nearest 100 mg. One ml of 100% methanol was added to each sample. The mixture was vortexed for five minutes, and the supernatant was collected after centrifugation at 12,000 rpm. The contents of 4-CPA and 4-CP in the supernatant were analyzed using HPLC-MS-MS, as described in the Appendix A. The experiment was replicated 4 times.

### 4.6. Transcriptome Analysis and qRT-PCR

Plants were treated with 0 (control) or 5 mg L^−1^ 4-CPA for 24 h and then harvested for RNA-sequencing. Total RNA was isolated using an RNeasy Plant Mini-Kit following the manufacturer’s instructions (Qiagen, #74903, Hilden, Germany). Construction of the Illumina library and the sequencing using the Illumina NovaSeq. 6000 platform were carried out using Novogene (Beijing, China, https://cn.novogene.com/, accessed on 24 September 2023). The data were analyzed as described in [50].

We validated 16 randomly selected genes (Appendix A) in the above samples using qRT-PCR to confirm the results of the transcriptome analysis. For each sample of total RNA, 1 mg of RNA was reverse transcribed using a PrimeScript RT-PCR Kit (TaKaRa, #R022A, Osaka, Japan). qRT-PCR was performed using a CFX96TM Real-Time System (Bio-Rad, Hercules, CA, USA) using iQ SYBR Green (Bio-Rad, #1708887, Hercules, CA, USA). A standard curve was generated for the quantitative determination of gene expression levels, as described in [15]. The rice actin gene OsACTIN (TIGR ID: Os03g50885) was used as an internal standard to normalize the cDNA concentrations.

### 4.7. Hydrogen Peroxide and POD Analysis

Plants were randomly assigned to 4-CPA, 4-CPA + WBPH, and control groups as stated above. The plant stems were harvested on the fourth day after starting the 4-CPA treatment. Each sample was ground in liquid nitrogen and weighed in two portions of 100 mg each for the following analyses. For the H_2_O_2_ analysis, a hydrogen peroxide content assay kit (Sangon Biotech, #D799774, Shanghai, China) was used following the manufacturer’s instructions. For the POD analysis, a peroxidase activity assay kit (Sangon Biotech, Shanghai, #D799592, China) was used following the manufacturer’s instructions. The analytical methods for H_2_O_2_ and POD are described in detail in the Appendix A. Each experiment was replicated 5 times.

### 4.8. Flavonoid, PA, and Lignin Analysis

For the flavonoid and lignin analyses, plants were randomly assigned to 4-CPA, 4-CPA + WBPH, and control groups, as described above. The plant stems were harvested on the fourth day after starting the 4-CPA treatment. Each sample was ground in liquid nitrogen, and 100 mg samples were weighed for the flavonoid analysis, as well as 300 mg samples for the liginin analysis. The flavonoid and lignin contents were extracted and quantified as described by [51,52] with minor modifications. See the Appendix A section for details. Each treatment was replicated four times.

For the PA analysis, plants were randomly assigned to 4-CPA and control groups as described above. The plant stems were harvested on the fourth day after starting the 4-CPA treatment. The extraction method of PAs was the same as that described for flavonoids. The PA analysis using HPLC-MS-MS was as described in [49]. Each treatment was replicated five times.

### 4.9. Location of Lignin-like Polymers

Plants were randomly assigned to 4-CPA, 4-CPA + WBPH, and control groups, as described above. After 4 days of treatment, 2 cm of the rice leaf sheaths were taken from the glass cage for paraffin sectioning. Cross-sectional sections of the rice leaf sheaths were stained with safranine and fast green, as described in [53]. The red fluorescence of lignin and lignin-like polymers in stained plant tissues was observed using laser confocal microscopy under the following excitation/emission set-up: λex = 561 nm, λem = 582–652 nm. Each treatment was replicated five times.

### 4.10. Greenhouse Bioassay

For the experiments used to determine stomach toxicity, we detected direct effects of 4-CPA and its metabolite 4-CP on the survival rate of WBPH nymphs. A total of 15 third-instar WBPH nymphs were kept in a two-way glass tube with two pieces of parafilm at each end. An artificial diet was located between the two pieces of parafilm. The specific formulation and rearing procedure of the artificial diet were presented as described by [54], with minor modifications. See the Appendix A section for details. The number of surviving nymphs was recorded for 4 consecutive days. Each treatment was replicated eight times.

To determine the survival rate and development time of the WBPH nymphs, a plant stem was placed in a 10 cm glass cage, into which 15 newly hatched nymphs were released. The rice plants had been treated with 5 mg L^−1^ 4-CPA for 0 or 4 days. A sponge was placed at the top of the cage to prevent the nymphs from escaping. The number of surviving nymphs was recorded for 9 consecutive days. From day 10, the number of fledged adults was recorded each day. This experiment was repeated eight times.

To determine nymph feeding preference, two plastic cups were planted with control and 4-CPA-treated (5 mg L^−1^ 4-CPA for 4 days) seedlings, and their stems were placed in the same glass cage in which 15 third-instar WBPH nymphs were released. A sponge was placed at the top of the cage to prevent the nymphs from escaping. The number of nymphs on the control and treated plants was recorded for 6 days. This experiment was repeated eight times.

### 4.11. Data Analysis

Differences in root mass, root length, stem strength, POD activity, and the levels of H_2_O_2_, lignin, and flavonoids for the different treatments were analyzed in SPSS software (ver. 17.0; SPSS Inc.,Chicago, IL, USA) using a one-way ANOVA. For the greenhouse experiments and PA analysis, differences in the control groups and treatment groups were analyzed in SPSS software using *t*-tests.

### 4.12. Data Availability

The RNA-seq data have been deposited in Sequence Read Archive (SRA) with accession PRJNA1018215.

## Figures and Tables

**Figure 1 ijms-24-15722-f001:**
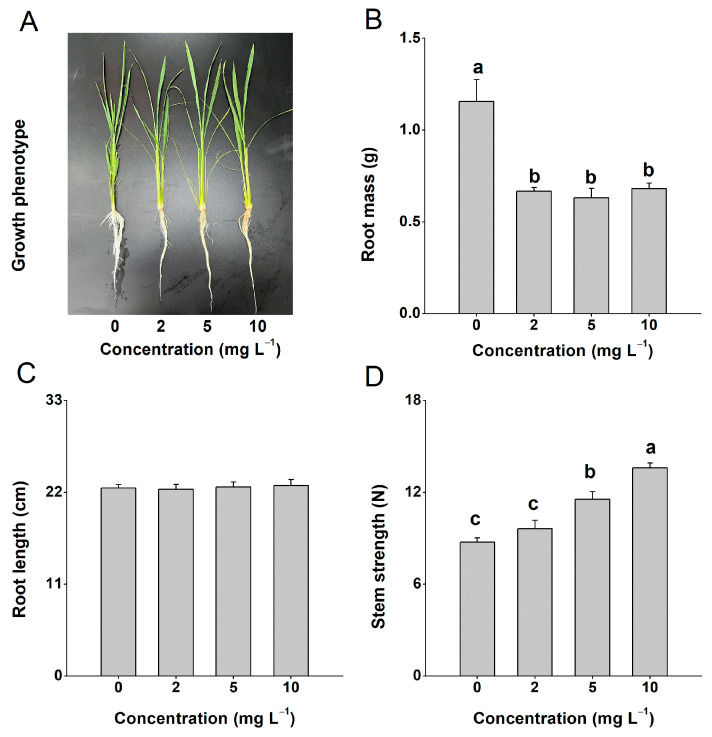
Four-CPA inhibits plant root growth. Growth phenotypes (**A**), root mass (**B**), root length (**C**), and stem strength (**D**) of plants treated with 0 to 10 mg L^−1^ 4-CPA for 10 d. Results are the average of six independent determinations + SE. Letters indicate significant differences among treatments (*p* < 0.05; Duncan’s multiple range test).

**Figure 2 ijms-24-15722-f002:**
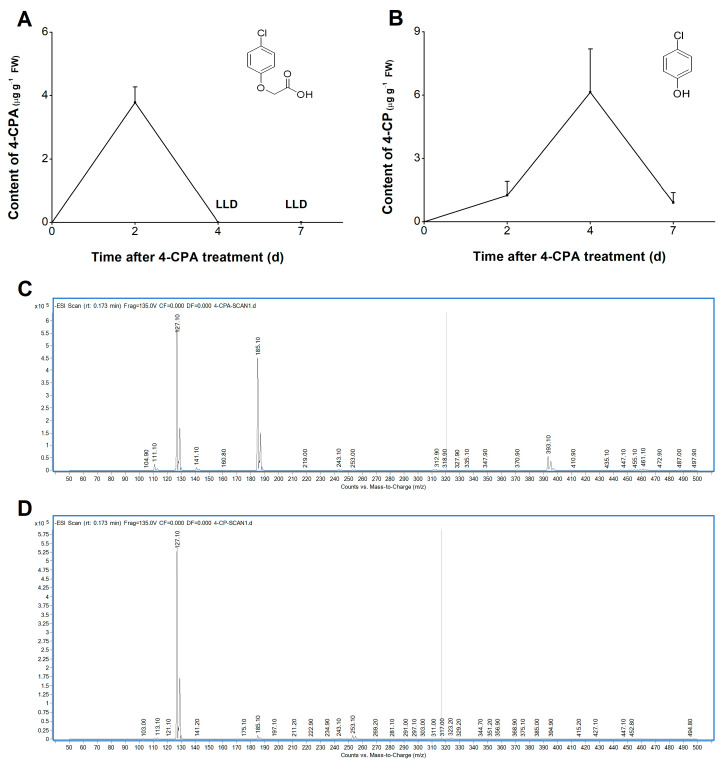
Four-CPA is degraded into 4-CP in plants. (**A**,**B**) Mean levels (+SE, *n* = 4) of 4-CPA (**A**) and 4-CP (**B**) in plant stems at 0, 2, 4, and 7 d after the start of 5 mg L^−1^ 4-CPA treatment. (**C**,**D**) Mass spectra of 4-CPA (**C**) and 4-CP (**D**) standards. LLD, lower limit of detection.

**Figure 3 ijms-24-15722-f003:**
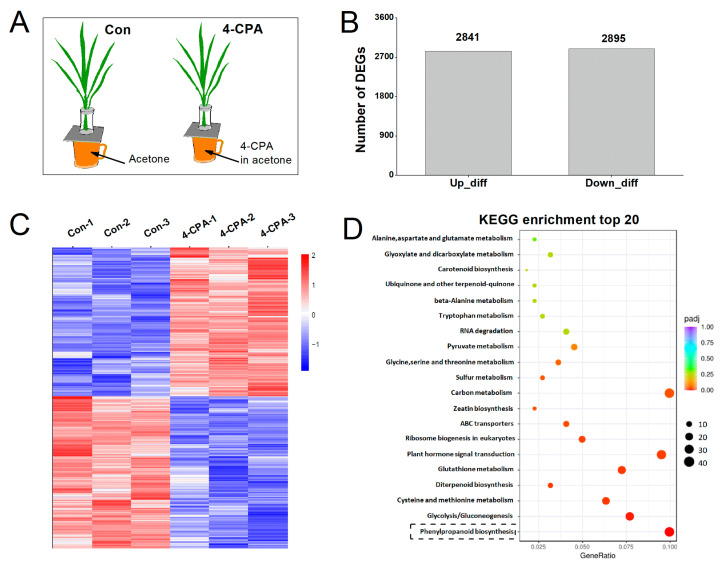
Differentially expressed genes (DEGs) in rice plants induced by 4-CPA compared to control plants (Con). (**A**) Treatment methods of the plants. (**B**) Number of upregulated and downregulated DEGs. (**C**) Heatmap of DEGs. (**D**) The top 20 KEGG enrichment pathways of the DEGs upregulated by 4-CPA. The black dashed box indicates the phenylpropanoid biosynthesis pathway, which was the metabolic pathway with the highest number of upregulated DEGs.

**Figure 4 ijms-24-15722-f004:**
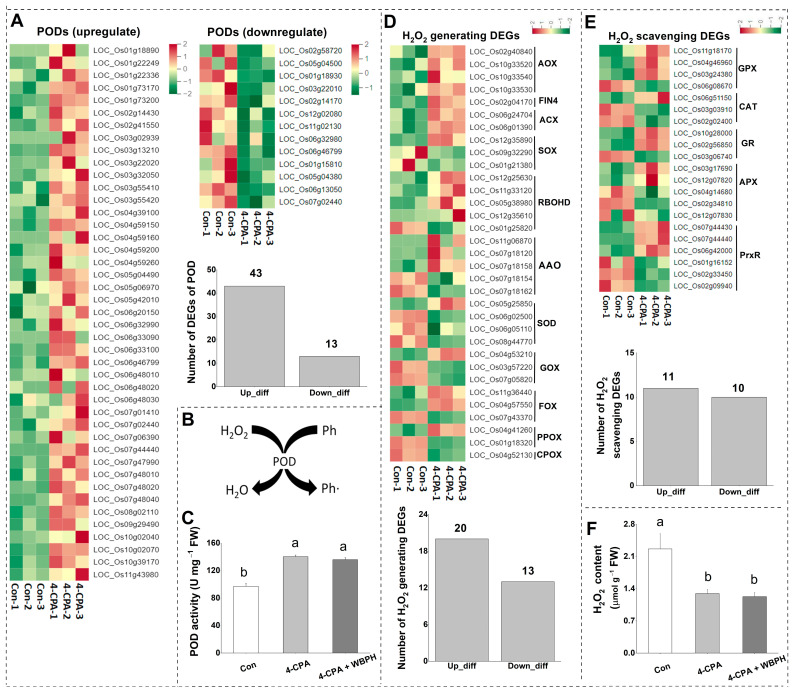
Four-CPA alters levels of hydrogen peroxide (H_2_O_2_) and activity of peroxidase (POD). (**A**) Heatmap and number of upregulated and downregulated PODs induced by 4-CPA. (**B**) The mechanism of POD. (**C**) Mean activity (+SE, *n* = 5) of POD in plants 4 d after the following treatments: 4-CPA, 4-CPA + WBPH, and control (Con). (**D**) Heatmap and number of DEGs involved in the H_2_O_2_ generating system induced by 4-CPA. (**E**) Heatmap and number of DEGs involved in the H_2_O_2_ scavenging (not including PODs) system induced by 4-CPA. (**F**) Mean levels (+SE, *n* = 5) of H_2_O_2_ in plants 4 d after the following treatments: 4-CPA, 4-CPA + WBPH, and control (Con). The numbers on the columns represent the number of DEGs. Letters indicate significant differences among treatment groups (*p* < 0.05; Duncan’s multiple range test). Ph, phenolic, Ph•, phenolic radical. H_2_O_2_-generating genes: AOX, alcohol oxidase; FIN4, L-aspartate oxidase; ACX, acyl-CoA oxidase; SOX, sarcosine oxidase; RBOHD, respiratory burst oxidase; AAO, aldehyde oxidase; SOD, superoxide dismutase; GOX, glycolate oxidase; FOX, flavin-containing amine oxidoreductase; PPOX, protoporphyrinogen oxidase; CPOX, coproporphyrinogen III oxidase. H_2_O_2_ scavenging genes: GPX, glutathione peroxidase; CAT, catalase; GR, glutathione reductase; APX, ascorbate peroxidase; PrxR, peroxiredoxin.

**Figure 5 ijms-24-15722-f005:**
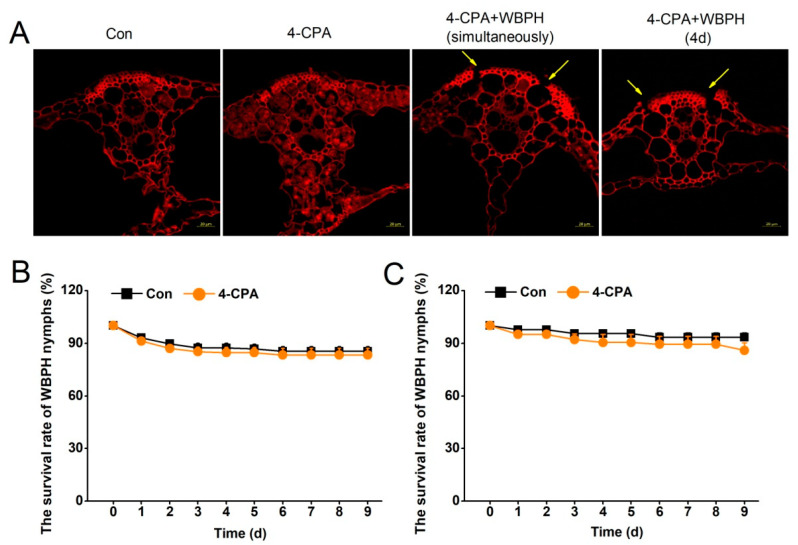
WBPH degrades 4-CPA-induced deposition of polymers. (**A**) Plants were divided randomly into 4 groups: Con, 4-CPA, 4-CPA + WBPH (simultaneously) (plants were treated with 4-CPA and infested with WBPH at the same time), and 4-CPA + WBPH (4 d) (plants were treated with 4-CPA for 4 days before infestation with WBPH). The leaf sheaths of the plants were harvested for paraffin sectioning after 4 days of the treatments. Paraffin sections were stained with safranine and fast green. A high intensity of red fluorescence corresponds to a high content of polymers. The yellow arrows indicate the WBPH feeding site. Scale bars = 20 µm. (**B**,**C**) Mean survival rate (+SE, *n* = 8) of 15 newly-hatched WBPH nymphs fed on plants that had been grown in nutrient solution with either 5 mg L^−1^ 4-CPA or without 4-CPA (Con) for 0 d (**B**), 4 d (**C**), or 1 to 9 days after infestation.

**Figure 6 ijms-24-15722-f006:**
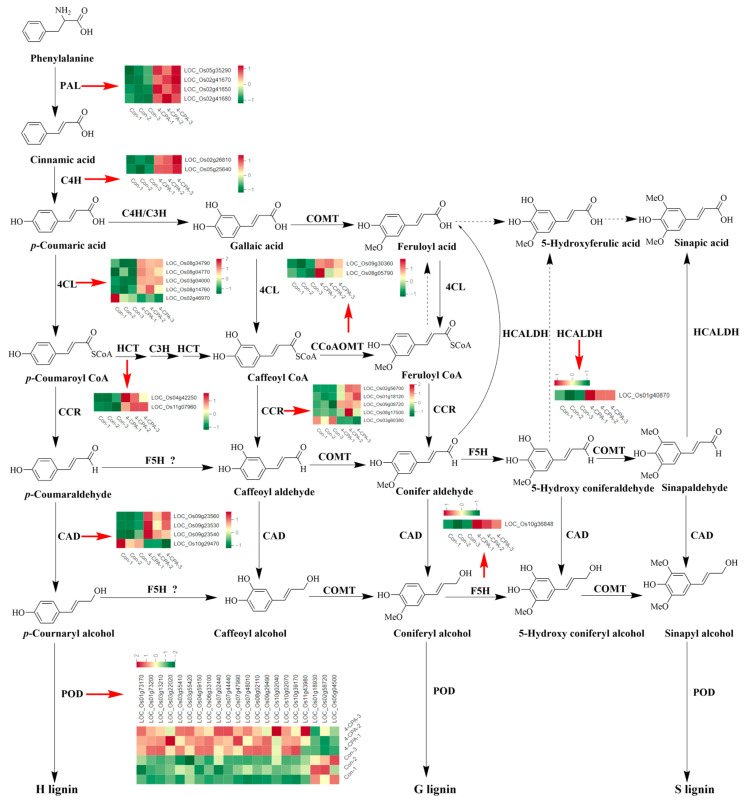
Expression profiles of genes involved in the phenylpropanoid biosynthesis pathway. The heat map provides differential gene expression profiles in control (Con-1, Con-2, Con-3) and 4-CPA-treated (4-CPA-1, 4-CPA-2, 4-CPA-3) plants. PAL, phenylalanine ammonia-lyase; C4H, trans-cinnamate 4-monooxygenase; 4CL, 4-coumarate-CoA ligase; CCR, cinnamoyl-CoA reductase; HCT, shikimate O-hydroxycinnamoyltransferase; C3H, coumarate-3-hydroxylase; CCOAOMT, caffeoyl-CoA O-methyltransferas; COMT, caffeic acid 3-O-methyltransferase; HCALDH, coniferyl-aldehyde dehydrogenase; F5H, ferulate-5-hydroxylase; CAD, cinnamyl-alcohol dehydrogenase; POD, peroxidase. The dot lines represented uncertain reaction processes in rice. The red arrows indicated the heatmap of each gene.

**Figure 7 ijms-24-15722-f007:**
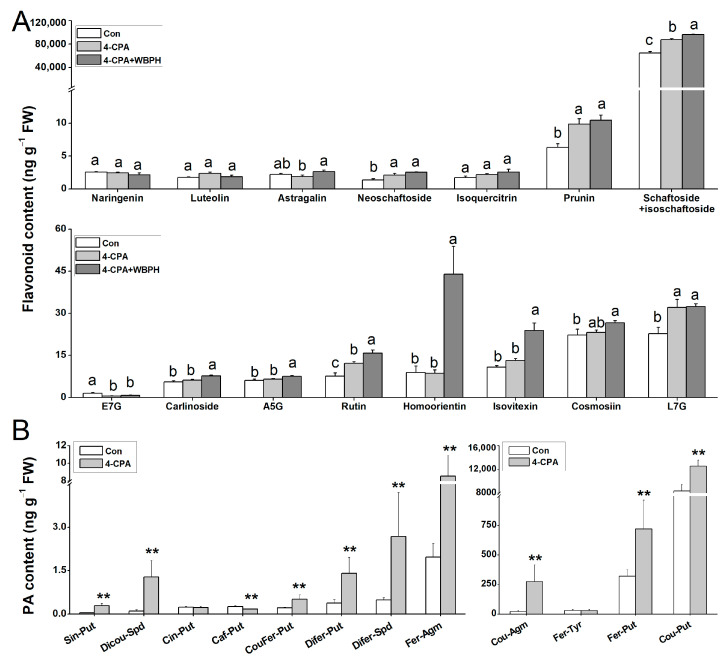
Four-CPA increases the levels of flavonoids and phenolamines (PAs). (**A**) Mean levels (+SE, *n* = 4) of flavonoids in plants 4 d after the following treatments: 4-CPA, 4-CPA + WBPH, and control (Con). (**B**) Mean levels (+SE, *n* = 5) of PAs in plants 4 d after the following treatments: 4-CPA and control (Con). Letters indicate significant differences among treatments (*p* < 0.05; Duncan’s multiple range test). Asterisks indicate significant differences between different treatments (** *p* < 0.05; *t*-test). E7G, Eriodictyol 7-0-glucoside; A5G, Apigenin 5-0-glucoside; L7G, Luteolin 7-0-glucoside; Cin-Put, N-cinnamoylputrescine; Caf-Put, N-caffeoylputrescine; Cou-Put, N-p-coumaroylputrescine; Fer-Put, N-feruloylputrescine; Cou-Agm, N-p-coumaroylagmatine; Sin-Put, N-sinapoylputrescine; Fer-Agm, feruloylagmatine; Fer-Tyr, N-feruloyltyramine; Sin-Agm, N-sinapoylagmatine; Dicou-Spd, N1,N10-dicoumaroylspermidine; CouFer-Put, N-p-coumaroyl-N′-feruloylputrescine; Difer-Spd, N1,N10-diferuloylspermidine; DiferPut, N,N’-diferuloylputrescine; Cou-Agm, N-p-coumaroylagmatine.

**Figure 8 ijms-24-15722-f008:**
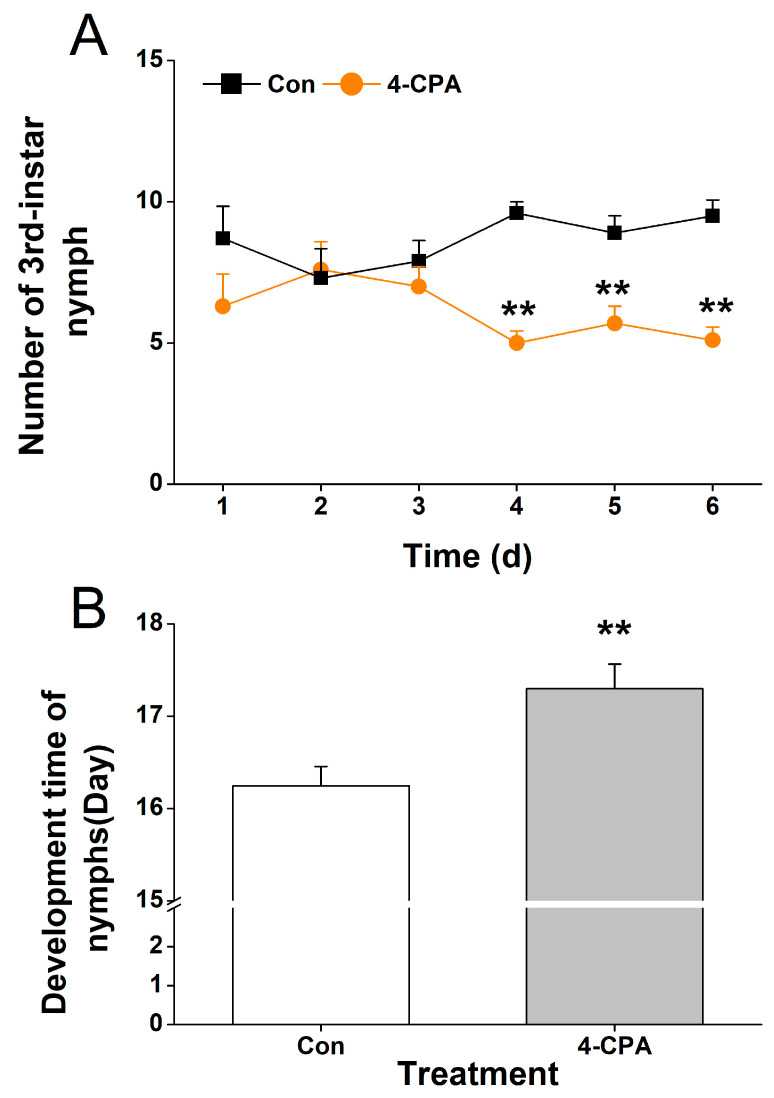
4-CPA treatment influences the performance of WBPH nymphs. (**A**) Mean number (+SE, *n* = 8) of third-instar nymphs fed on plants that had been treated with either 5 mg L^−1^ 4-CPA or without 4-CPA (Con) for 4 d. (**B**) Mean development time (+SE, *n* = 60) of nymphs that fed on plants that had been treated with either 5 mg L^−1^ 4-CPA or without 4-CPA (Con) for 4 d. Asterisks indicate significant differences between different treatments (** *p* < 0.05; *t*-test).

**Figure 9 ijms-24-15722-f009:**
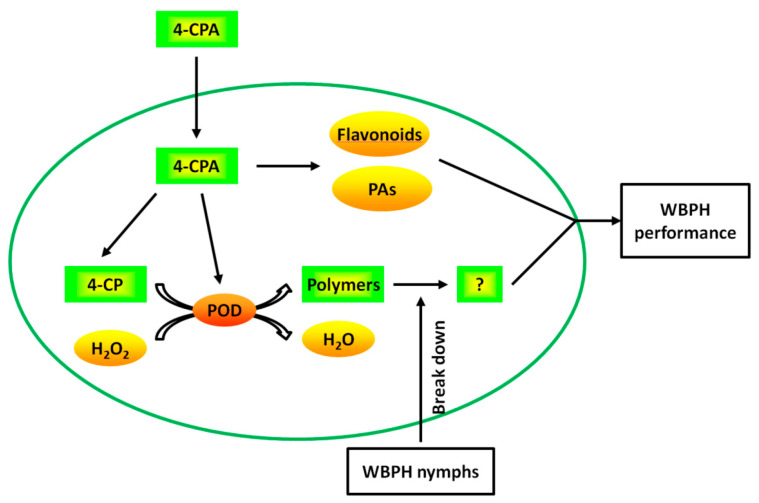
The proposed model of the mechanism of 4-CPA. After being absorbed by plant roots, 4-CPA undergoes rapid side-chain degradation to yield 4-CP. Four-CPA treatment induces the activity of POD. POD then catalyzes 4-CP and possibly other phenolics in plants to form polymers using H_2_O_2_ as the electron acceptor. The polymers are degraded by WBPH nymphs. Meanwhile, 4-CPA increases the levels of flavonoids and PAs. The increased flavonoids and PAs, together with the degradation products of the polymers, affect WBPH performance.

## Data Availability

The RNA-seq data have been deposited in Sequence Read Archive (SRA) with accession PRJNA1018215. Other data are contained within the article or in the Appendix A.

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
