# Peer review of "Four-Chlorophenoxyacetic Acid Treatment Induces the Defense Resistance of Rice to White-Backed Planthopper Sogatella furcifera"

_ijms, 2023, doi:10.3390/ijms242115722_

Round 1

Reviewer 1 Report

Comments and Suggestions for Authors
It is an exciting scientific report with promising results. However, little is known about the environmental toxicity of 4-Chlorophenoxyacetic acid.  I recommend: It would be helpful if the authors raise this concern at the end of the Discussion:   4-Chlorphenoxyactic acid is a toxic agent, but the possible new research on pathways/compounds could resolve this concern.

Reviewer 2 Report

Comments and Suggestions for Authors

Manuscript entitled “Four-chlorophenoxyacetic Acid Treatment Induces the Defense Resistance of Rice to White-backed Planthopper Sogatella furcifera” by Wanwan Wang et al. this article analyzes different aspects of 4-CPA in rice defenses to the White-backed Planthopper.

The work is interesting but, in my opinion, the manuscript presents some issues mainly in the Figures, Materials and Methods, and References, and, as well as other things, that need to be checked to make this work suitable for publication.

1. Main issues:

1.1. Regarding Figures.

Figure 2. The letters and numbers in panels C and D are too small and cannot be read. if they can't be improved, make them bigger as supplementary material.

Figure 2. Why are there two black boxes in panel D? The vertical words in the two black boxes are too small and not visible

Line 137. Please correct “induced by 4-CPA.” with “induced by 4-CPA compared to control (Con).”

Figure 4. The letters and numbers are too small and cannot be read properly.

Figure 6. The font and number size is at the limit, if possible increase them a little bit.

1.2. Regarding Tables.

1.3. Regarding section Materials and Methods

I must point to the authors that, according to the Instructions for Authors section of this journal, Full experimental details must be provided so that the results can be reproduced.  The Materials and Methods section of this manuscript is not detailed enough for other authors to repeat some of the experiments.

4.1. Please, indicate the light intensity

4.5. Please explain more detailed how the extraction with methanol was done (volume, there was filtering, etc.).

4.6. A supplementary table with the primers is missing.  Selected genes and their primers used for qRT-PCR. Include one similar to the S7 table in your article “Induction of defense in cereals by 4-fluorophenoxyacetic acid suppresses insect pest populations and increases crop yields in the field”

4.9. Line 459. The wavelength is missing.

1.4. Regarding section References

The references list has numerous format mistakes. According to the “Instructions for Authors” (Journal Articles - Author 1, A.B.; Author 2, C.D. Title of the article. Abbreviated Journal Name Year, Volume, page range.) the references list has numerous format mistakes.

The correct form of Abbreviated Journal Name is for example in reference 4 “Proc. Natl. Acad. Sci. U. S. A.” and not “Proc Natl Acad Sci U S A”. In reference 11 “Biosci. Biotech. Bioch.” and not “Biosci Biotech Bioch”. Please revise all the text.

In the references 13, 21, etc. the Journal name is not abbreviated. Please revise all the text.

The volume should be in italics.

Revise the italics (e.g. Line 758 Please correct “cis” and “Arabidopsis” with “cis” and “Arabidopsis”, Line 587 “Catharanthus roseus”. Revise the entire manuscript).

The DOI of each reference must be removed.

Almost all the words in the title begin with capital letters. Many of them should not be in that format (reference 24).

Please revise all the references

2. Additional comments

All text. Please correct “Figure.” with “Figure”

Line 5. Please correct “1, 2,*” with “1,2,*”

Line 36. Please check “such as volicitininceptincaeliferinsbruchins36 β-glucosidase”. the commas have a strange format.

Line 53. Please correct “[14,16,17].JA” with “[14,16,17]. JA”

Line 54. Please correct “JA-Ile” with “jasmonoyl-isoleucine (JA-Ile)”

Line 72. Please correct “2, 4-Dichlorophenoxyacetic” with “2,4-Dichlorophenoxyacetic”

Line 73. Please correct “(2, 4-D).” with “(2,4-D). ”

Line 91. Please correct “(Figure. 1A).” with “(Figure 1, A and B). ”

Line 122. Please correct “(Con)” with “(control; Con).”

Line 125. Please correct “4242 of 8323” with “4,242 of 8,323”

Line 126. Please correct “4081” with “4,081”

Line 128. Please correct “data. (Table S1).” with “data (Table S1).”

Line 131. Please correct “5045 of 8323” with “5,045 of 8,323”

Line 135. what does 75 refer to? 75% of the DEGs o 75 pathways.

Line 143. Please correct “H2O2were” with “H2O2 were”

Line 276. Please correct “(Figure.1A and B).” with “(Figure 1, A and B).”

Line 295. Please correct “[peroxidase/ H2O2/phenolic] system” with “[peroxidase/H2O2/phenolic] system”

Line 360. Please correct “Fig” with “Figure”

Lines 437 and 446. Please correct “100mg” with “100 mg”

Line 450. Please correct “, Plants” with “, plants”

Lines 418, 470 and 475. Please correct “mg L−1” with “mg L−1

Line 482. Please correct “H2O2” with “H2O2

Line 488. Add the accession.

Author Contribution

According to the “Instructions for Authors” :The following statements should be used "Conceptualization, X.X. and Y.Y.; Methodology, X.X.; Software, X.X.; Validation, X.X., Y.Y. and Z.Z.; Formal Analysis, X.X.; Investigation, X.X.; Resources, X.X.; Data Curation, X.X.; Writing – Original Draft Preparation, X.X.; Writing – Review & Editing, X.X.; Visualization, X.X.; Supervision, X.X.; Project Administration, X.X.; Funding Acquisition, Y.Y.”,

Please put this section in the correct formatting

Reviewer 3 Report

Comments and Suggestions for Authors

The presented article is an interesting study of how chemical elicitors can increase plant defence against herbivorous insects. The study focused on the 4-chlorophenoxyacetic acid (4-CPA), which can induce chemical defences to protect rice plants from planthoppers ( Sogatella furcifera).

The paper is well prepared; however, some deficiencies have to corrected.

Abstract: lines 27-28. authors write, "These results indicate that 4-CPA has a potential to be used as a chemical elicitor for insect pest management". --This context is not correct because 4-CPA inhibits the growth of rice.

The authors contradict themselves because other elicitors inhibiting plant growth have not been approved.

  See discussion- Line 273-275: An important reason why chemical elicitors have not yet been used commercially is their negative effect on plant growth. As a plant growth regulator, 4-CPA significantly inhibited the growth of rice roots at a concentration ranging from 2 to 10 mg L-1 (Figure.1A and B).

Introduction: In lines 77-86, the study results data are not necessary, and the same data are also included in the abstract and summary of the discussion.

Moreover, this place needs the aim of the studies and the hypothesis.

Why such research, and on what basis was it conducted?

The hypothesis, presented by the authors, is posted to the result (line 173) and the discussion (line 286). These elements should be reported in the introduction.

The results: 

Line: 91: Authors write: "Exogenous application of 4-CPA inhibited plant growth (Figure. 1A): however, the study is limited to the root data, but the above-ground part of the plant is not evaluated. 

Lines: 181-182: should be added to this part - After what time did these polymers disappear, and how long did these planthoppers larvae feed?

Lines 184-185: It needs to be understood. Is there a result of this study or another study? Is the sentence cited after other authors?  

Line 256: authors write: "And the development time of WBPH nymphs fed on 4-CPA-treated plants was longer than that of nymphs fed on control plants (Figure. 8B)". - but the question is, what is the difference in the development time of these WBPH nymphs? The data in Figure 8b shows only one day difference, so the differences are insignificant for using the 4-CPA.

 Discussion: line 370: premature statement because that study includes only one species of planthopper, not all insects pests.

Other comments  in the PDF

Round 2

Reviewer 2 Report

Comments and Suggestions for Authors

Manuscript entitled “Four-chlorophenoxyacetic Acid Treatment Induces the Defense Resistance of Rice to White-backed Planthopper Sogatella furcifera” by Wanwan Wang et al. this article analyzes different aspects of 4-CPA in rice defenses to the White-backed Planthopper.

The authors have made a considerable improvement in the manuscript but there are still things to modify to make this work suitable for publication

1. Main issues:

1.1. Regarding Figures.

Figures and Tables should be number according to its citation in the main text.

Figure S3 (in line 109) is the Figure S1

Figure S4 (in line 109) is the Figure S2

Figure S2 (in line 116, 364) is the Figure S3

Figure S1 (in lines 239, 299) is the Figure S4

The authors should modify the number also in the section Supplementary Materials (Lines 500 to 503) and in the document with the Supporting Information

1.2. Regarding Materials and Methods.

The authors have omitted section 4.12, which is very important for other researchers to have access to these data. Please add it and indicate the accession number.

I must point to the authors that, according to the Instructions for Authors section of this journal in the section “Supplementary Materials, Data Deposit and Software Source Code”:

Deposition of Sequences and Expression Data

New sequence information must be deposited to the appropriate database prior to submission of the manuscript. Accession numbers provided by the database should be included in the submitted manuscript. Manuscripts will not be published until the accession number is provided.

New high throughput sequencing (HTS) datasets (RNA-seq, ChIP-Seq, degradome analysis, …) must be deposited either in the GEO database or in the NCBI’s Sequence Read Archive (SRA).

2. Additional comments

Line 124. Please correct “conrol” with “control”

Line 496. Please correct “H2O2” with “H2O2

Line 546. Please correct “Philosophical Transactions of the Royal Society B Biological Sciences” with “Philos. Trans. R. Soc. B-Biol. Sci.”. Line 561. Please correct “Plant, Cell & Environment” with “Plant Cell Environ.”. Please revise all the text (line 563, 578, 579, etc.)

Round 3

Reviewer 2 Report

Comments and Suggestions for Authors

The authors have done a great job.  I congratulate authors on the research and its findings.

Author Response

Thank you for recognising our work.